# Hybrid topological photonic crystals

Yanan Wang[1,9], Hai-Xiao Wang [2,9] ✉, Li Liang[1], Weiwei Zhu[3], Longzhen Fan[1], Zhi-Kang Lin[4], Feifei Li[1], Xiao Zhang[1], Pi-Gang Luan[5], Yin Poo [1] ✉, Jian-Hua Jiang[4,6] ✉ & Guang-Yu Guo [7,8] ✉

Topologically protected photonic edge states offer unprecedented robust propagation of photons that are promising for waveguiding, lasing, and quantum information processing. Here, we report on the discovery of a class of hybrid topological photonic crystals that host simultaneously quantum anomalous Hall and valley Hall phases in different photonic band gaps. The underlying hybrid topology manifests itself in the edge channels as the coexistence of the dual-band chiral edge states and unbalanced valley Hall edge states. We experimentally realize the hybrid topological photonic crystal, unveil its unique topological transitions, and verify its unconventional dual-band gap topological edge states using pump-probe techniques. Furthermore, we demonstrate that the dual-band photonic topological edge channels can serve as frequency-multiplexing devices that function as both beam splitters and combiners. Our study unveils hybrid topological insulators as an exotic topological state of photons as well as a promising route toward future applications in topological photonics.

Topological photonics is an interdisciplinary field at the interface between photonics and topological physics, which has greatly fertilized both fields in the past decade[1–5]. Hallmark topological phenomena such as unidirectional backscattering-immune photonic edge states were discovered with analog to quantum anomalous Hall (QAH) insulators in time-reversal broken photonic systems[6–15]. Furthermore, diversified topological photonic phases, including photonic Floquet topological insulators[16–20], photonic quantum spin Hall[21–27], and photonic valley Hall (VH) insulators[28–31] are observed and find remarkable applications in integrated[32–36], nonlinear[37–40], and quantum photonics[41–43]. For instance, owing to the robust topologically protected edge channels, topological insulator lasers can outperform conventional lasers[44,45]. From the fundamental aspect, a unique feature of photonic systems is their nonequilibrium nature, i.e., photons can be excited, transported, and detected at any desired frequency. The nonequilibrium nature of photons opens new possibilities and gives

rise to unconventional opportunities in topological photonics. Most strikingly, the nonequilibrium nature of photons enables the discovery of many topological phenomena at room temperature, even when photon energy is significantly smaller than the thermal fluctuation energy at room temperatures[6–45]. Furthermore, owing to the nonequilibrium nature, photonic topological phenomena can involve multiple band gaps, leading to dual-band topology[38,46–51] and even non-Abelian topology[52–55]. Such multi-band-gap photonic topological states can enable multiplexing of topological edge modes[46–51] and edge-enhanced resonant nonlinear photonic effects[38].

However, most studies focus on multi-gap topological photonic systems with the same topological class[38,47,49,50], while that with distinct topological classes has not yet been realized. It was reported that the topological valley and pseudo-spin edge states can coexist in a Kekulé photonic system[48] and a composite photonic crystal[51], which seem to be multi-gap topological photonic systems with different topological

[1]School of Electronic Science and Engineering, Nanjing University, Nanjing 210093, China. [2]School of Physical Science and Technology, Guangxi Normal University, Guilin 541004, China. [3]College of Physics and Optoelectronic Engineering, Ocean University of China, Qingdao 266100, China. [4]School of Physical Science and Technology, & Collaborative Innovation Center of Suzhou Nano Science and Technology, Soochow University, Suzhou 215006, China. [5]Department of Optics and Photonics, National Central University, Jhongli 32001, Taiwan. [6]Suzhou Institute for Advanced Reseach, University of Science and Technology of China, Suzhou 215123, China. [7]Department of Physics, National Taiwan University, Taipei 10617, Taiwan. [8]Physics Division, National Center for Theoretical Sciences, Taipei 10617, Taiwan. [9]These authors contributed equally: Yanan Wang, Hai-Xiao Wang. ✉e-mail: hxwang0216@gmail.com; ypoo@nju.edu.cn; joejhjiang@hotmail.com; gyguo@phys.ntu.edu.tw

classes. However, from the symmetry consideration of photonic topological phases in two dimensions, the photonic quantum spin Hall insulator phase is protected by concurrent parity ($\mathcal{P}$) and time-reversal ($\mathcal{T}$) symmetries. In comparison, the photonic VH insulator phase requires the breaking of $\mathcal{P}$, while the photonic QAH insulator phase requires the breaking of $\mathcal{T}$. Therefore, the photonic quantum spin Hall insulator phase is incompatible with the latter two. In contrast, the photonic VH insulator and QAH insulator phases are compatible with each other. Therefore, it is possible in principle to have multi-band topology of the QAH and VH types in a single photonic system if both $\mathcal{P}$ and $\mathcal{T}$ are broken which, however, has not yet been realized.

Here, we report on the realization and discovery of an exotic photonic topological phase that exhibits simultaneously QAH and VH topology (see Fig. 1a, b) in a single photonic crystal system dubbed as hybrid topological photonic crystals (HTPCs). An intriguing feature of HTPCs is that the band topology can be switched from one type to another different type by changing just the frequency of photons. In other words, distinct topological phenomena can be realized in the same photonic system to enable multiplexing photonic topological edge transport with very different properties (see Fig. 1c, d). The HTPCs give rise to the simultaneous emergence of the unidirectional chiral edge states and unbalanced valley edge states in different band gaps. Due to the breaking of both the $\mathcal{P}$ and $\mathcal{T}$ symmetries, the unbalanced valley edge states have different absolute group velocities for different valleys, which are distinct from the existing valley edge states that have exactly opposite group velocities for different valleys. Moreover, here the photonic VH phases have large valley Chern numbers and are characterized by unconventional topological transitions characterized by an unpaired quadratic point at the $K$ or $K'$ point. At the edge boundaries, the photonic VH phases studied here have

multiple valley-polarized edge states. These unconventional properties give promise to novel topological phenomena and valuable applications in photonics such as advanced wave filters and frequency-multiplexing devices that function as both beam splitters and combiners.

## Results

### Design of the HTPC

The HTPC here forms a hexagonal lattice with the lattice constant $a = 21$ mm, as illustrated in Fig. 2a. Each unit cell includes a Y-shaped gyromagnetic rod with three identical arms, of which the width is $W = 1.76$ mm and the length is $L = 3.89$ mm (see Methods for more material parameters). The HTPC is cladded by metallic plates from above and below to form two-dimensional photonic systems dominated by transverse-magnetic modes. The spatial symmetry and topological phases of the HTPC are controlled by the rotation angle $\theta$. If one starts from the case with $\theta = 0°$ and zero external magnetic field, the band structure has some paired Dirac points at the $K(K')$ point and a quadratic point at the $\Gamma$ point that are protected by both $C_{3v}$ and $\mathcal{T}$ symmetries (see Supplementary Fig. 1). By applying an external magnetic field, all Dirac points are gapped and topological band gaps (indicated by the light-blue blocks in Fig. 2b) are formed. For convenience, we term the band gap between the third and fourth bands (the fourth and fifth bands) as gap II (III) and focus on the frequency range within the blue box in Fig. 2b henceforth. The calculated photonic Chern numbers indicate that both gaps II and III are Chern gaps, i.e., photonic analogs of the QAH phase (see Supplementary Note 1 for details). Next, by increasing $\theta$, both gaps II and III at the $K$ valley are reduced while those at the $K'$ valley are enlarged. At $\theta = 9.5°$, gap III closes at the $K$ point (see Fig. 2c), yielding an unpaired quadratic point

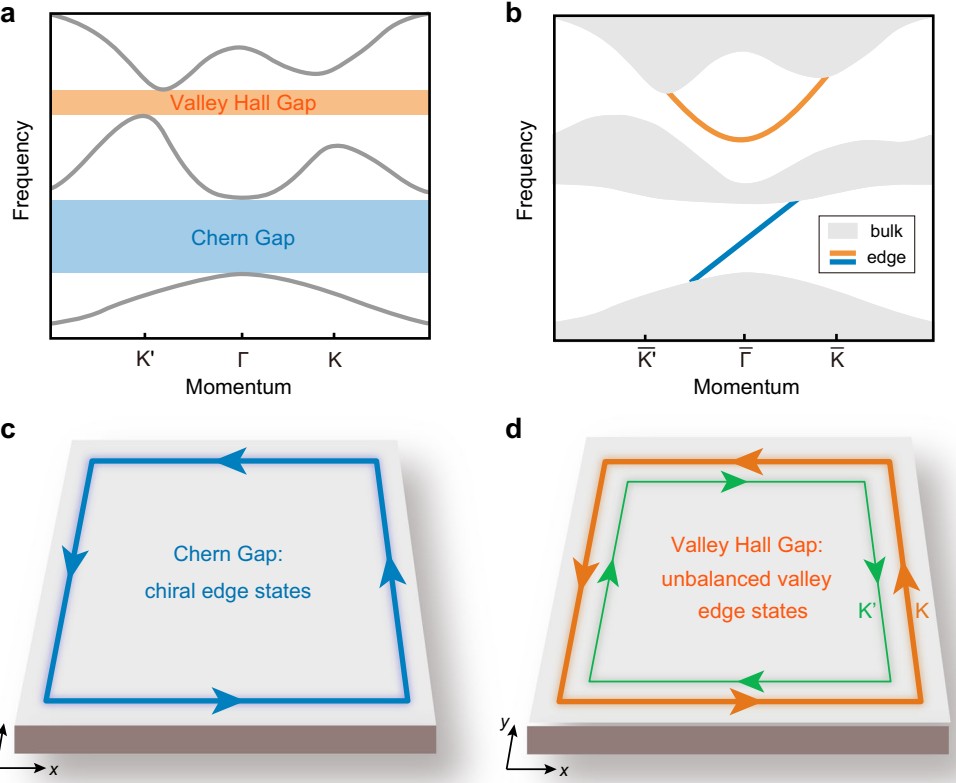

**Fig. 1 | A hybrid topological system with distinct topology in the different band gaps. a** Bulk bands with a Chern gap and a valley Hall gap. The Chern gap is characterized by an integer Chern number, while the valley Hall gap is characterized by valley Chern numbers. **b–d** The resultant edge states in the system. **b** Illustration of the edge states in different band gaps. **c** The Chern gap hosts chiral edge states. **d** The valley Hall gap hosts unbalanced valley edge states where the absolute group velocity of the valley edge state around the $K$ valley is different from that around the $K'$ valley.

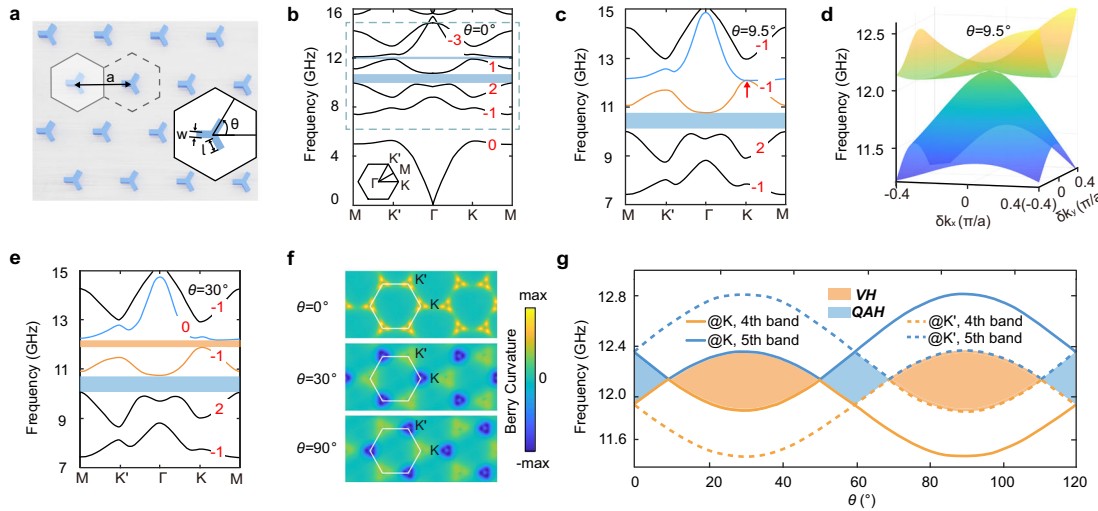

**Fig. 2 | Hybrid topological photonic crystal (HTPC). a** Schematic of an HTPC consisting of Y-shaped gyromagnetic rods, where the lattice constant $a = 21$ mm, and the length (width) of the three identical arms is $W = 1.76$ mm ($L = 3.89$ mm). $\theta$ is a tunable rotation angle. **b** Photonic band structure of the HTPC for $\theta = 0°$ under the external effective magnetic field of 700 Oe. Blue zones indicate the band gaps with finite Chern numbers. The red number at each band indicates the Chern number of the band. Inset: the first Brillouin zone. **c** Photonic band structure of the HTPC for $\theta = 9.5°$. The unpaired quadratic point at the $K$ point is indicated by the red arrow. **d** The quadratic dispersion around Dirac point. **e** Photonic band structure of the HTPC for $\theta = 30°$ (HTPC1). The orange zone indicates the VH band gap with a zero Chern number. **f** Calculated Berry curvature of all bands below gap III for $\theta = 0°$ (upper panel), $\theta = 30°$ (middle panel), and $\theta = 90°$ (lower panel), respectively. For these cases, the Chern number of gap III is 2, 0, 0, separately. The Brillouin zone is labeled by the hexagon. **g** Topological phase diagram versus the rotation angle $\theta$, where the light blue (orange) area refers to the QAH (VH) phase.

at a finite momentum (its dispersion is shown in Fig. 2d) as both the $C_{3v}$ and $\mathcal{T}$ symmetries are broken. Here, the unpaired quadratic point serves as an unconventional topological transition between the QAH and VH phases in gap III (in comparison, similar transitions in the Haldane model are through unpaired Dirac points[56,57]). Such an unpaired quadratic Dirac point can be gapped by further increasing $\theta$. Figure 2e presents the band structure of HTPC with $\theta = 30°$, in which gap III is of the VH phase. Remarkably, here we emphasize that gapping a quadratic Dirac point gives rise to an integer valley Chern number, in contrast to the common perception that a valley Chern number takes a value of $\pm\frac{1}{2}$ when gapping a Dirac point at a finite momentum[28,29,31,33–36,47–51]. This is confirmed via two approaches: Berry curvature calculations (see Fig. 2f) and the analytical theory (see Supplementary Note 2 for details).

The full phase diagram of the HTPC is shown in Fig. 2f when the rotation angle $\theta$ is tuned from 0° to 120° (i.e., the minimal periodicity considering the three-fold rotation symmetry of the HTPC). During the whole tuning process, the Chern number of gap II remains as $C_{II} = 1$. From the phase diagram, we find that the topological transitions of gap III take place at $\theta = 60° \times n \pm 9.5°$ ($n$ is an integer), where the unpaired quadratic point appears at the $K$ or $K'$ point. To reveal the nature of the topological transition in gap III, we present a **k · p** theory for the effective Hamiltonian of the photonic bands around the $K$ and $K'$ valleys. We denote the Bloch states at the $K$ ($K'$) point for the fourth and fifth bands, respectively, as $|4,K\rangle$ ($|4,K'\rangle$) and $|5,K\rangle$ ($|5,K'\rangle$). Using the basis $(|4,K\rangle, |4,K'\rangle, |5,K\rangle, |5,K'\rangle)^T$, the **k · p** Hamiltonian can be written as

$$H(\vec{k}) = A_Q\hat{\tau}_0\left[\left(k_x^2 - k_y^2\right)\hat{\sigma}_x - 2k_xk_y\hat{\sigma}_z\right] + B_Qk^2\hat{\tau}_0\hat{\sigma}_0 + (m_T\hat{\tau}_0 - m_v\hat{\tau}_z)\hat{\sigma}_z,$$

(1)

where **k** $= (k_x, k_y)$ is the displacement of the wavevector relative to the $K$ or $K'$ point. $A_Q$ and $B_Q$ are the band parameters of the quadratic point. Here, $\hat{\sigma}_i$ and $\hat{\tau}_i (i = x,y,z)$ are the Pauli matrices acting on the orbital and valley subspaces, $m_V$ and $m_T$ are the mass terms induced by breaking the $C_{3v}$ (through rotation) and $\mathcal{T}$ (through an external magnetic field) symmetries, separately. When $m_T = m_V = 0$, i.e., in the

case with $\theta = 0°$ and zero external magnetic field, there are two quadratic points located at the $K$ and $K'$ points (see Supplementary Fig. 1), respectively. By breaking the symmetries, a band gap can be open whose magnitude is proportional to $|m_T - m_V|$ at the $K$ valley and $|m_T + m_V|$ at the $K'$ valley.

Starting from Eq. (1), by integrating the Berry curvature, one finds that the valley Chern numbers (i.e., the Chern number of a specific valley) are $C_K = -sgn(m_T - m_V)$ and $C_{K'} = -sgn(m_T + m_V)$ (see Supplementary Note 2). Here, the external magnetic field gives $m_T < 0$, while the rotation operation gives $m_V < 0$ for $\theta \in (0°, 60°)$ and $m_V > 0$ for $\theta \in (60°, 120°)$. Therefore, at $\theta = 0°$, $C_K = C_{K'} = 1$, and the total Chern number of gap III is $C_{III} = 2$. We find that $m_V$ first decrease with $\theta$. At $\theta = 9.5°$, $m_V = m_T$, and the gap at the $K$ point is closed, leading to an unpaired quadratic point. With the further decrease of $m_V$, the gap reopens but $C_K$ switches sign, leading to $C_K = -C_{K'} = -1$, i.e., a VH phase with large valley Chern number[58,59]. After $\theta = 30°$, $m_V$ starts to increase with $\theta$. $m_V$ comes back to $m_T$ at $\theta = 50.5°$, leading to a transition back to the QAH phase. With further increase of $\theta$, similar transitions take place at the $K'$ point while the $K$ point remains gapped in the phase diagram. The difference here is that the valley Chern number is reversed in the VH phase with $\theta > 60°$, i.e., $C_K = -C_{K'} = 1$.

## Observation of multiplexing edge states with distinct topological origins

We now test the multiplexing edge states depicted in Fig. 1. For simplicity, we denote the HTPCs with $\theta = 30°$ and $\theta = 90°$, respectively, as HTPC1 and HTPC2. First, a ribbon-shaped HTPC2 supercell terminated by perfect electric conductors (see Fig. 3a and Methods for more simulation details) is used to calculate the edge spectrum. As shown in Fig. 3b, two edge branches emerge in gap II whose typical electric field patterns (labeled as A and B) and their Poynting vector distributions are shown in Fig. 3c. These are the one-way photonic chiral edge states due to the QAH topology in gap II: the group velocities of edge states at opposite edge boundaries are of opposite signs. This unidirectional feature is also confirmed by the energy flow (Poynting vector) distributions. Meanwhile, in gap III, unbalanced valley Hall edge states emerge. Here, the electric field patterns (labeled as C and D) and the

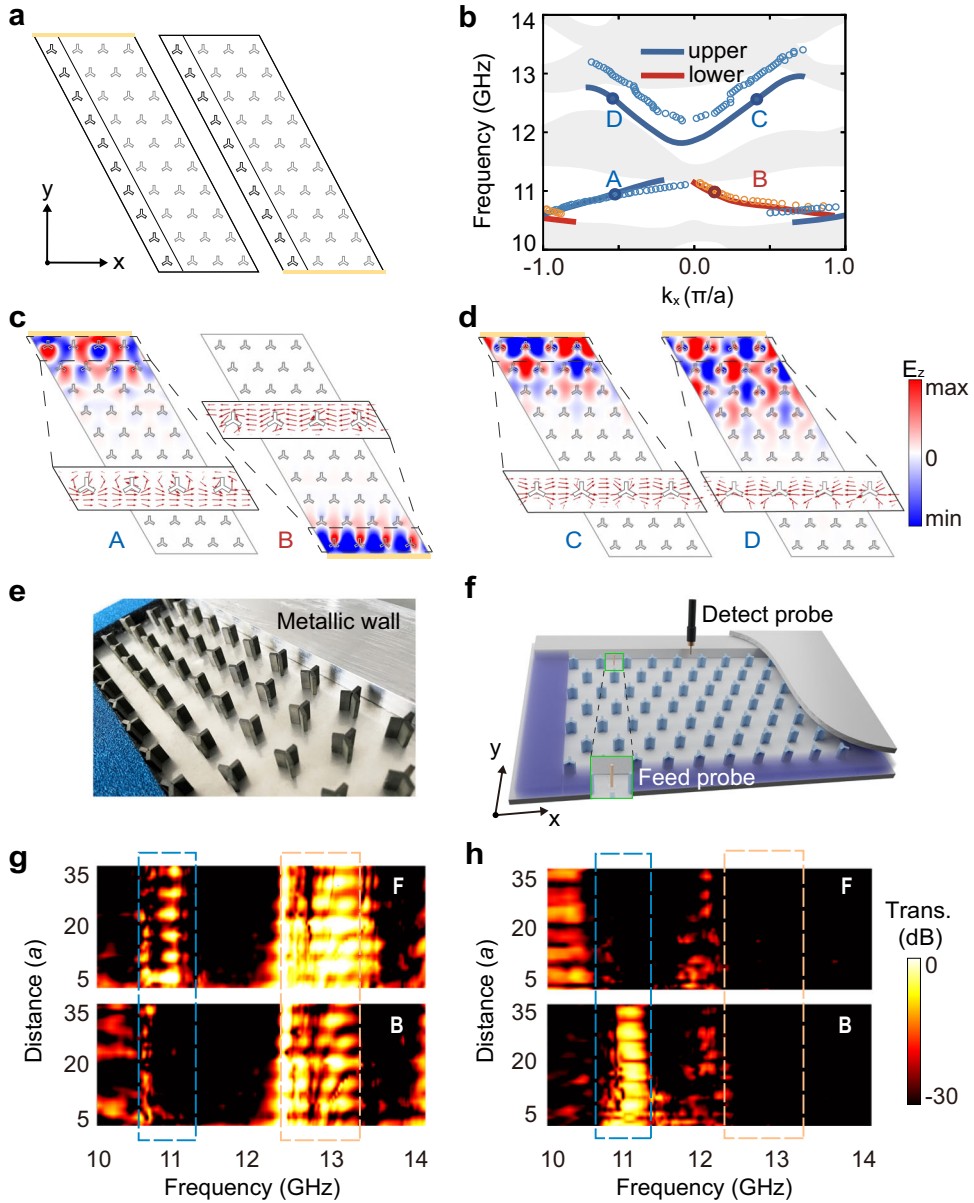

**Fig. 3 | Multiplexing edge states with distinct topological origins. a** Schematic of the supercell consisting of ten HTPC2 unit cells terminated by perfect electric conductors, where the supercell is expanded into four periods in the x-direction for easy viewing. **b** The simulated (solid lines) and experimental (empty circles) edge dispersions. The gray area refers to the bulk states and the colored edge states are observed in both gaps II and III. **c**, **d** The electric field pattern of **c** the chiral edge states A and B and **d** the unbalanced valley edge states C and D. Insets: zoom in on the Poynting vector along the boundary. **e** A close view of the sample bounded with a metallic cladding. **f** The experimental setups for transmission measurement. **g**, **h** The forward (labeled with "F") and backward (labeled with "B") transmissions as functions of frequency and the distance between the source and the detection points along the upper (**g**) and the lower (**h**) edge channels, respectively. The blue dashed boxes indicate the nonreciprocal propagation of chiral edge states in gap II. The orange dashed box in **g** indicates bidirectional propagation of the valley edge states in the upper edge channel, while in **h** indicates the absence of edge states in the lower edge channel.

Poynting vector distributions (see Fig. 3d) are quite different from the chiral edge states in gap II. Figure 3d indicates that the edge states localized at the upper edge have opposite energy flows. Besides, the edge states in gap III at the same edge boundary have both positive and negative group velocities, indicating that they are not unidirectional edge states. Furthermore, there is no edge state in the lower edge boundary in gap III since the HTPC with rigid boundary does not strictly contribute a bulk-edge correspondence and the edge dispersions also depend on the boundary configuration, making it boundary-configurable valley edge states[60,61] (see Supplementary Note 3 for more details).

We then experimentally verify the coexistence of multiple edge states in gaps II and III by implementing the transmission measurement

in a finite-sized sample, as depicted in Fig. 3e, f. The experimentally measured edge dispersions (indicated by the empty circles in Fig. 3b, also see Methods for more experiment details) are in good agreement with those from the finite-element simulation. To unveil the topological behavior of the edge states in gaps II and III, we present both the forward (labeled with "F") and the backward (labeled with "B") transmission spectra for photon flow along the upper and lower edge channels in Fig. 3g, h, respectively. For the frequency window ranging from 10.61 to 11.25 GHz (indicated by the blue dashed box, also see Supplementary Note 4 for the identification of the bulk gap), it is seen that the nonreciprocal photon flows exist in both the upper and lower edge channels, indicating the existence of the unidirectional edge states. For the higher frequency window ranging from 12.34 to

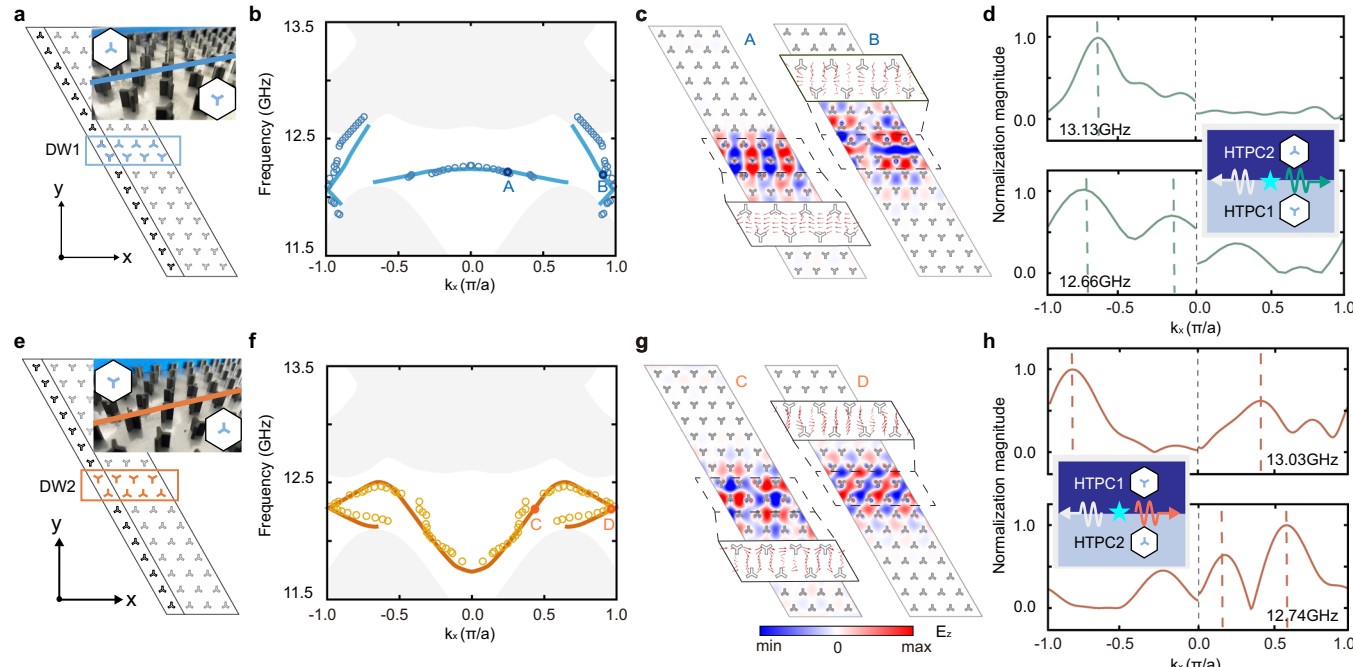

**Fig. 4 | Valley edge states in the domain wall systems consisting of HTPCs.**
**a**, **e** Illustration of domain wall formed by HTPC1 and HTPC2. In **a** DW1 refers to the configuration with HTPC1 above HTPC2. In **e** DW2 refers to the configuration with HTPC1 below HTPC2. Note that the supercells are extended into four periods in the *x*-direction to show the direction of the edge channel. Insets: Photographs of the experimental samples. **b**, **f** The simulated (solid lines) and shifted experimental (solid circles) valley edge dispersions for DW1 (**b**) and DW2 (**f**). The gray area refers to the bulk states. **c**, **g** The simulated electric field patterns of the edge states **c** in DW1: *A* and *B* and **g** in DW2: *C* and *D*. Insets: the Poynting vector profiles of the edge states around the domain walls. **d**, **h** Forward transmission spectra versus wave-vectors for **d** DW1 with the frequency of 13.13 GHz (upper panel) and 12.66 GHz (lower panel), **h** DW2 with the frequency of 13.03 GHz (upper panel) and 12.74 GHz (lower panel). Insets: schematics of the measurement setups.

13.17 GHz (indicated by the orange dashed box), the forward and backward transmissions show the bidirectional propagation of the valley-polarized edge states in the upper edge channel. Meanwhile, the vanished forward and backward transmissions indicate the absence of edge states in the lower edge channel, being consistent with the simulated results in Fig. 3b.

Next, we study domain wall systems formed by the HTPC1 and HTPC2 with opposite valley Chern numbers, where the HTPC2 on the top of HTPC1 is termed DW1, and that on the bottom of HTPC1 is termed DW2, as schematically shown in Fig. 4a, e, respectively (see Methods for more simulation details). Because these two HTPCs have identical Chern numbers of the gap II, no topological edge states can survive at DW1 or DW2. In contrast, it is expected that two valley edge states emerge at DW1 (DW2) since the absolute value of the valley-contrasting Chern number (i.e., the difference in the valley Chern number across the domain wall) is 2. The eigen spectrum of the DW1 and DW2 are shown in Fig. 4b, f, respectively, where the gray regions and lines represent the projections of bulk bands and the dispersions of valley edge states. Both spectra indicate that two pairs of valley edge states within gap III emerge at the domain walls. Note that for DW1, two valley edge states only survive in a narrow frequency window. Despite it, the valley edge states exhibit valley-momentum locking behavior, which can be checked by the typical electric field patterns and their Poynting vector distributions (labeled as *A* and *B* for DW, and *C* and *D* for DW2) in Fig. 4c, g.

To confirm the valley-polarized edge states, we implement transmission measurements in the finite-sized samples (see the insets of Fig. 4a, e, also see Methods for more experiment details). It is seen that the measured valley edge state dispersions (indicated by the empty circles in Fig. 4b, f) are in good agreement with the simulation results. To further illustrate the valley edge dispersions, we consider the forward transmission spectra with serval typical frequencies, as displayed in Fig. 4d, h. For the DW1, it is seen that there are two peaks

with negative wavevector in the transmission spectrum with a frequency of 12.66 GHz (lower panel in Fig. 4d), identifying that DW1 supports two distinct valley edge modes. However, when increasing the frequency to 13.13 GHz (upper panel in Fig. 4d), only one peak with a negative wavevector is observed, indicating only one valley-polarized edge mode survives in DW1. In parallel, for the DW2, it is seen that there are two peaks in transmission spectra with frequencies of 12.74 GHz (lower panel in Fig. 4h) and 13.03 GHz (upper panel in Fig. 4h), respectively, indicating that DW2 support two valley edge modes. However, the phase velocities (wavevectors) of the edge modes with a frequency of 13.03 GHz exhibit opposite signs, while that of 12.74 GHz hosts the same signs.

**Frequency-dependent topological routing via HTPCs**
The dual-band gap edge states with different boundary configurations revealed above could be useful for designing frequency-dependent topological routings. As depicted in Fig. 5a, a three-port topological routing consisting of HTPC1 and HTPC2 are cladded by metallic walls (see Methods for more simulation details). When an excitation source, of which the frequency is within gap II, is placed at the upper boundary, it is expected that the wave can only propagate from P1 to P3 since these two HTPCs have identical Chern numbers in gap II and thus no edge states can survive in the sloped interface (indicated by the orange arrows in Fig. 5a). In contrast, no edge states exist at the upper boundary of HTPC1 when the operating frequency is within gap III, making the wave propagate from P1 to P4 (along the Z-shape route, illustrated by the green arrows). The above topological routing effect is further demonstrated by the simulated transmission spectrum (see Fig. 5b) and electric field distributions (see Fig. 5c, d). For the straight edge channel (from P1 to P3), the transmission is nearly unity within gap II, while experiencing a decrease within gap III (light orange area). In contrast, for the Z-shape edge channel (from P1 to P4), there exists

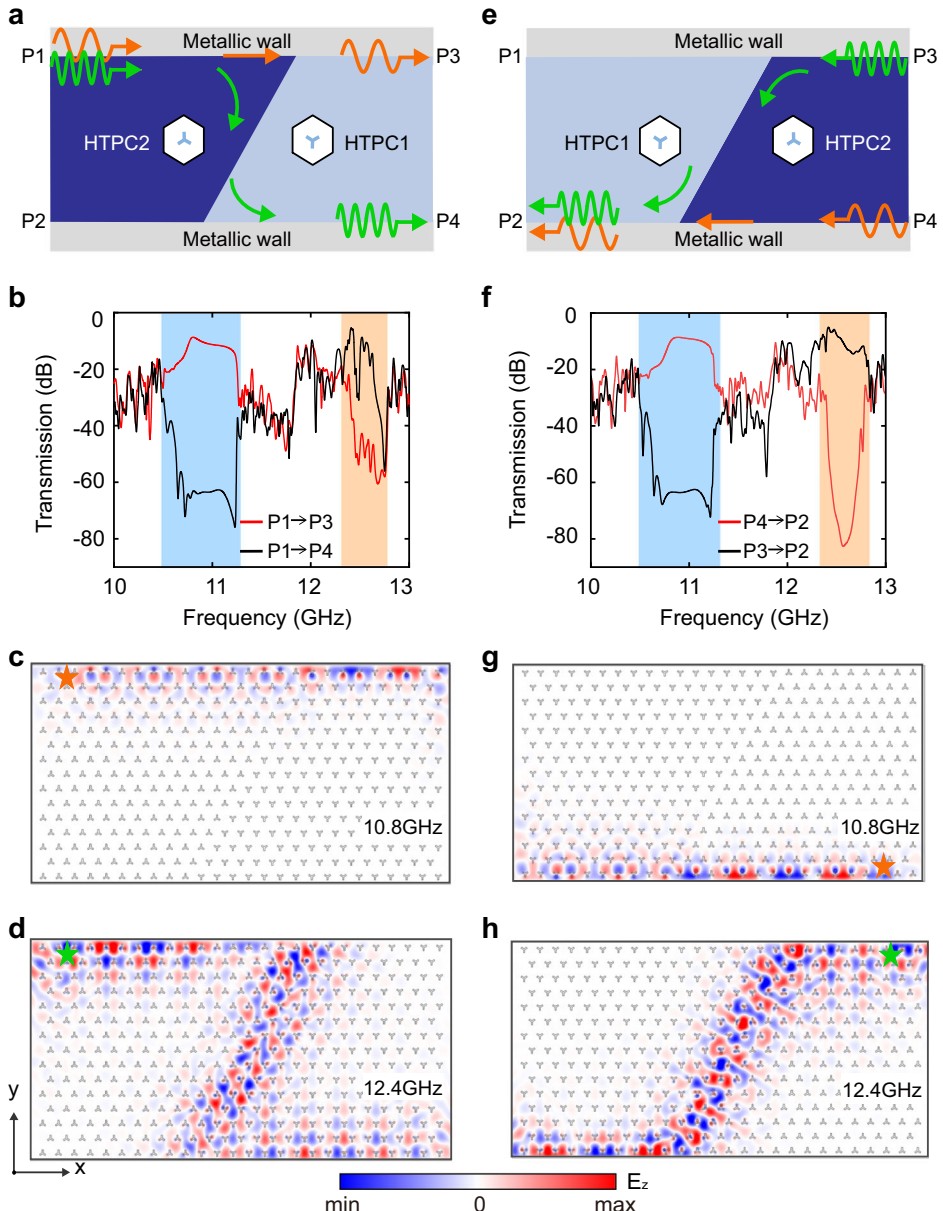

**Fig. 5 | Frequency-dependent topological routings based on HTPC structures.**
**a**, **e** Illustration of frequency-dependent topological routing based on different configurations: **a** Z-shaped boundary with HTPC1 on the left and **e** Z-shaped boundary with HTPC2 on the left. In **a** the edge waves with a lower (higher) frequency in gap II (III) propagate from P1 to P3 (P4), as indicated by the orange (green) arrows. In **e**, the edge waves with a lower (higher) frequency in gap II (III) propagate from P4 (P3) to P2, as indicated by the orange (green) arrows. **b**, **f** The simulated transmission spectra for (**b**) and (**f**) that confirm the frequency-selective topological routing in (**a**) and (**e**): i.e., gap II is dominated by the straight edge

channels whereas gap III is dominated by the Z-shaped and inverted Z-shaped edge channels. The light yellow (green) region refers to gap II (III). **c**, **g** Typical simulated electric field distributions of the straight edge channel in gap II for the configurations in (**a**) and (**e**). Here, the electromagnetic waves are excited by point sources (the orange stars) with a frequency of 10.8 GHz. **d**, **h** Typical simulated electric field distributions of the Z-shaped and inverted Z-shaped edge channels in gap III for the configurations in (**a**) and (**e**), which are excited by point sources (the green stars) with a frequency of 12.4 GHz.

an obvious drop within gap II (light blue area) while the transmission remains unity with gap III. A typical simulated electric field distribution at 10.8 GHz in Fig. 5c, of which the frequency is within the gap II, shows that the electromagnetic waves are well confined at the perfect electric conductor boundary and propagate along the straight edge channel. Meanwhile, another typical simulated electric field distribution at 12.4 GHz in Fig. 5d, of which the frequency is within gap II, indicates that the electromagnetic waves propagate unidirectionally along the Z-shape edge channel.

In addition, exchanging the configurations of HTPCs yields another three-port topological routing (see Fig. 5e). At this time, the

electromagnetic waves cannot propagate from P1 to P4 (P3) due to the nonreciprocity character induced by the breaking of $\mathcal{T}$ symmetry. However, one can still realize a three-port topological routing by placing an emitter at either P3 or P4 and a receiver at P2. As illustrated in Fig. 5e, the receiver placed at P2 can accept wave signals either from P3 with high frequency (along the inverted Z-shaped edge channel, indicated by the green arrows) or P4 with lower frequency (along the straight edge channel, indicated by the orange). Such a proposal is further demonstrated by the simulated transmission spectrum in Fig. 5f. It is seen that gap II is dominated by the straight edge channel, whereas gap III is dominated by the inverted Z-shaped edge channel,

similar to that in Fig. 5e. Furthermore, we also provide two typical simulated electric field distributions at 10.8 and 12.4 GHz in Fig. 5g, h, respectively. Indeed, it is seen that the electromagnetic waves are mainly localized along straight (inverted Z-shaped) edge channels when an excitation source placed at P4 (P3) with a lower (higher) frequency is excited.

## Discussions

We unveil an exotic topological phase of photons: HTPCs which have distinct topology in adjacent band gaps as enabled by the breaking of both $\mathcal{P}$ and $\mathcal{T}$ symmetries. Here, the two photonic band gaps exhibit the QAH and VH topology, respectively, as characterized by distinct edge states and topological numbers. In addition to its fundamental value, the discovery of HTPCs may also benefit future applications in topological photonics. For instance, HTPCs with multiplexing in edge channels can enable the simultaneous realization of beam splitting[62–64] and beam combining for photonic edge transport. It may also enable highly efficient topological photon filtering due to the distinct edge modes in different photonic band gaps. The photonic wave multiplexing in the edge channels of HTPCs offers a promising future for topological wave manipulation in photonics. Finally, we remark that although the HTPCs are demonstrated using gyromagnetic photonic crystals, the main results here can be further extended to Floquet photonic topological insulators[16,17,65–69] (see Supplementary Note 6 for our proposed model of Floquet hybrid topological photonic crystal with intriguing multi-gap topology). Therefore, the concept of hybrid topology with distinct topological classes can be generalized to optical frequencies via Floquet photonic topological insulators. Moreover, when nonlinear effects are considered, interactions between the topological edge states of distinct nature can enable nonlinear switch of photonic flows in the topological edge channels, e.g., switching from the unidirectional photonic edge flow to the bidirectional photonic edge flow via the nonlinear optical effects (see Supplementary Note 7 for a concrete example that demonstrates such an effect via simulations), which enriches the degree of freedom for the manipulation of photon propagation in topological edge channels. These discoveries unveil exotic phenomena and possibilities in the field of topological photonics.

## Methods

### Materials

All the gyromagnetic rods used in the experiment are made of yttrium iron garnet (YIG), a typical magneto-optical material in the microwave regime to break the $\mathcal{T}$ symmetry. The relative permittivity is about 15.29 at X-band. Typically, under fully transverse saturated magnetization, the YIG ferrite processes strong anisotropy corresponding to tensor permeability expressed as follows

$$\mu = \begin{pmatrix} \mu_r & -i\kappa & 0 \\ i\kappa & \mu_r & 0 \\ 0 & 0 & 1 \end{pmatrix} \qquad (2)$$

where

$$\mu_r = 1 + \frac{\omega_m(\omega_0 + i\alpha\omega)}{(\omega_0 + i\alpha\omega)^2 - \omega^2}, \qquad (3a)$$

$$\kappa = \frac{\omega_m\omega}{(\omega_0 + i\alpha\omega)^2 - \omega^2}, \qquad (3b)$$

and $\omega_m = 4\pi\gamma M_s$ is the characteristic frequency with gyromagnetic ratio $\gamma = 2.8$ MHz/Oe and saturation magnetization $4\pi M_s = 1884$ Gaussian. $\omega_0 = \gamma H_0$ is the resonant frequency proportional to the external magnetic field $H_0$. $\omega$ is the operating angular frequency.

### Simulations

All simulations in this paper are implemented with the radio frequency module of COMSOL Multiphysics. To obtain the bulk bands, the boundaries of the primitive cell are set to be periodic. The band structures in Fig. 3 are calculated using a supercell that consists of ten HTPC2 terminated by perfect electric conductors, while another supercell consisting of eight HTPC1 and eight HTPC2 cladded by perfect electric conductors are employed to calculate the valley edge dispersions in domain wall systems in Fig. 4. The stimulated transmission spectra and the electric field patterns in Fig. 5 are calculated by exciting a point source with scanning frequencies.

### Experiments

Two samples are fabricated in our experiments. A sample consisting of HTPC2 with 5 × 12 unit cells is designed to demonstrate the coexistence of multiple edge states in gap II and III in Fig. 3e. The other sample is composed of HTPC1 and HTPC2 with 8 × 12 unit cells, as shown in the insets of Fig. 4a, e, respectively. The experimental setups for transmission measurement are illustrated in Fig. 3f, where both fixed feed probe and slidable detect probe are inserted in the interface between HTPCs and a metallic wall. The whole structure is sandwiched between two metallic paralleled plates with three sides surrounded by electromagnetic absorbers to mimic a two-dimensional environment. The external magnetic field is applied with $H_0 = 900$Oe (the effective magnetic field is 700 Oe after considering the demagnetization). The measured edge dispersions in experiments utilize the Fourier-transformed field scan method. Note that the measured dispersion has been shifted downwards by 0.5 GHz to account for the air layer between the sample and upper plate of the parallel plate waveguide (see Supplementary Note 5 for the original dispersion of the valley edge states).

## Data availability

All data were available in the manuscript and the Supplementary Information. Additional information is available from the corresponding authors through proper request.

## Code availability

We use the commercial software COMSOL MULTIPHYSICS to perform electromagnetic wave simulations and eigenstates calculations. All related codes can be built using the instructions in the Method section.

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

## Acknowledgements

Y.P. thanks the support of the National Key Research and Development Program of China (Grant No. 2022YFA1203500). Y.W., L.L., L.F., F.L., X.Z., and Y.P. are supported by the National Natural Science Foundation of China (Grant No. 62171215 and No. 62001212), STP of Jiangsu Province (BK20201249), the Priority Academic Program Development of Jiangsu Higher Education Institutions and Jiangsu Provincial Key Laboratory of Advanced Manipulating Technique of Electromagnetic Wave, the young scientific and technological talents promotion project of Jiangsu Province and Zhongying Scholarship. H.-X.W. is supported by the National Natural Science Foundation of China (Grant No. No. 11904060) and Natural Science Foundation of Guangxi Province (Grant No. 2023GXNSFAA026048), J.-H.J. and Z.-K.L. are supported by the National Natural Science Foundation of China (Grant No. 12074281 and No. 12125504), and the Jiangsu Province Specially-Appointed Professor Funding. G.-Y.G. is supported by the National Science and Technology Council and the National Center for Theoretical Sciences in Taiwan.

## Author contributions

H.-X.W., Y.P., and G.-Y.G. initiated the project. H.-X.W., J.-H.J., and Y.P. guided the research. H.-X.W., L.L., W.Z., J.-H.J., and P.-G.L. established the theory. Y.W., H.-X.W., and L.L. performed the numerical calculations and simulations. Y.W., L.F., F.L., X.Z., and Y.P. designed and achieved the experimental set-up and the measurements. Y.W., H.-X.W., L.L., L.F., and Z.-K.L. drew the figures. All the authors contributed to the discussions of the results and the manuscript preparation. H.-X.W., J.-H.J., Y.P., and G.Y.G. wrote the manuscript and the Supplementary Information.

## Competing interests

The authors declare no competing interests.
