## [Peer Review File · Nature Communications]

REVIEWER COMMENTS

Reviewer #1 (Remarks to the Author):

In the paper “Hybrid topological photonic crystals” by Y. Wang et al. the authors demonstrate that photonic phases with different topologies, namely quantum anomalous Hall and valley Hall phases, can exist in the same photonic crystal. This is so because the quantum anomalous Hall phase requires the breaking of time-reversal (\mathcal{T}) symmetry whereas the valley Hall phase requires the breaking of the parity (\mathcal{P}) symmetry. Therefore, it is no surprising at all that in a system in which both these symmetries are broken both topological phases can be created and as such the main result demonstrated by the authors is expected and to some extent trivial. Moreover, the authors do not explain why it would be important to have both topological phases in the same photonic crystal. This is an important issue because the photonic system considered by the authors is linear and therefore the two phases with different topologies (topological edge modes) do not couple and therefore they do not interact. As such, the fact that they occur simultaneously in the same photonic system has no practical relevance. It would have been much more interesting to see that the authors consider the case of a nonlinear photonic crystal, so that the optical modes with different topologies can interact. Finally, another serious drawback of the paper is that the physics of the new structure is demonstrated at microwave frequencies, using magnetic materials. Therefore, these results cannot be easily extended to optical frequencies, and as such the results reported here would have very little practical impact. Given all these comments, I recommend that the paper is not published in Nature Communications.

Reviewer #2 (Remarks to the Author):

In the manuscript by Yanan Wang, et al., the authors propose and experimentally investigate a hybrid topological photonic system which simultaneously exhibits anomalous Hall effect (AHE) type of topology with broken time-reversal symmetry and valley-Hall (VH) type of topology with broken inversion symmetry. As such, the system is characterized by two topological invariants, and allows multiplexing via two kinds of topological boundary states. To the best of my knowledge, this is first experimental study which would show such multispectral robust topological transport via different topological states in different band gaps. The paper is also very well-written, experiments are fascinating, and I only suggest to slightly tune down hype in the abstract (e.g. “rich photonic edge transport measurements”, which sounds really too much since measurements cannot be rich). Another point of criticism I would like to raise is that the authors do not cite earlier works which too studies hybrid topological systems, including combining AHE and VH effects (theoretical work by Ni et al., Science Advances, 7, 2, (2021)), as well as another experimental work combining spin-Hall (SH) and VH (Kang et al., Nature Commun. 9, 3029 (2018)) based on earlier SHE experimental work (Cheng et al., Nature Materials 15, 542 (2016)). Both works study the effect of two mass terms, topological transitions between different topological phases,

and evaluate proper hybrid topological invariants. I note, however, that these earlier works, while very relevant, do not undermine novelty of the manuscript under review, because here heterogeneous modes in topologically distinct and spectrally separated bandgaps are studied for the first time. Nonetheless, earlier works must be given proper credit in the manuscript with appropriate mentions and citations. Provided these minor suggestions are implemented, I strongly recommend this work for publication in Nature Communications. I think this is an important result which shows many novel aspects which can be achieved in hybrid topological photonic systems, including higher values of hybrid topological invariants, heterogeneous multispectral topological transport, etc. I am confident that this paper will be of significant interest to the broad research community and Nature Communications readership.

Reviewer #3 (Remarks to the Author):

The manuscript 'Hybrid topological photonic crystals' by Y. Wang, H.-X. Wang, et al. reports theoretical designs and experimental demonstrations of a 2D T- and P-broken photonic crystal featuring gaps of distinct nontrivial topology, namely Chern and valley Hall insulating gaps. The design concept and approach is interesting and clearly displays the intended phenomenon. Similarly, the experimental demonstrations in the microwave-regime back up the theoretical predictions very convincingly. Even separately for Chern and valley Hall edge states, the demonstrations are among the clearest that I've seen. Finally, the authors suggest a possible routing application for their design.

The work and methodology is sound and explained clearly.

I am unsure how impactful the idea of a dual-gap-topology ultimately could be in practice, especially of valley Hall type topology. Nevertheless, the demonstration is beautiful and amounts to a particularly clear illustration of the potential of "abundant" co-existing topological phenomena in photonic crystals. The design idea is also neat - particularly, in using a structural rotation angle to realize the desired dual-gap topology.

Overall, despite my doubts about the impact and practical utility of dual-gap-topology, I believe the clarity of the experimental results and presentational quality is sufficiently high that the paper will have clear impact in the community of topological photonics.

Provided the authors can fix or reply to the few points I raise below, I would recommend publication in Nature Communications.

Main points:

- The authors claim in the introduction (and, to a degree, in the abstract) that 'It is unclear whether photonic band gaps in a single system can have distinct topologies'. Technically, this is true in that no one has explicitly shown this - but conceptually, it is obviously possible. Of course, the existence of practical designs is not a given. Nevertheless, I suggest the authors temper this statement to reflect that the real uncertainty here is not whether it is possible to have distinct band topologies in distinct gaps, but just whether practical designs of such dual-gap topology can exist and how we should think about engineering them.

- Can the authors comment on the near-constant frequency shift between theory and experiments, especially pronounced for the valley edge state (e.g., Fig. 3b and Supplementary Fig. 6)?

- The authors note a selection of earlier work [43-48] on dual band topology but do not give a clear exposition of how they differ from the present work. Can the authors expand on these differences, e.g. in the Introduction? For instance, [44-47] are suggested to have identical topology in distinct bands - in what sense are they then dual band topology? Perhaps especially relevant is a clarification relative to Ref. 48.

Minor points/suggestions:

* Figure 1b,c,e: to facilitate the discussion of how the Chern number of band 4 changes under variations of θ , the authors should label what the Chern number is of band 5. Since the change of Chern number of band 4 is achieved by an inversion with band 5, the sum of Chern numbers of the two should be invariant wrt. θ before after 9.5° . On this point, the Chern number of band 4 changes by 2 (from 1 to -1) under the band inversion: nominally, this is surprising, since a single band inversion usually only contributes a change of 1. The authors should explain why this is possible (is it e.g., related to the unpaired nature of the quadratic degeneracy at 9.5° ?).

* Figure 1(a-b): Although the figures are conceptual and hence do not require explicit frequency units or values, it would be helpful if the authors could include relevant labels on the momentum axis (since the valley Hall effect involves a specific choice of K or K' valley); even just an indication of the zero-momentum would be helpful.

* 'Unbalanced valley edge states' are mentioned a few times without immediate explanation. I suggest the authors either give a brief description of what is meant by 'unbalanced' at the first mention, or hold off on introducing the modifier until it is explained. (As I understand it from the text, different valleys get different absolute group velocities)

* p. 4, bottom: The authors mention 'paired Dirac points': without reference to the SI, I could not guess that this meant a 'doublet of Dirac points', one at K and K' (as required by time-reversal symmetry). I suggest the authors be slightly more elaborate here.

* Eq. (1): The first two terms ostensibly feature an implicit Kronecker product with τ_0 : this should be made explicit for clarity's sake.

* The authors refer to 'boundary-configured valley edge states' and reference [57,58]. I don't believe this is widely known terminology: a brief explanation of the term's meaning is appropriate here.

* Fig. 3b: it would be nice to clarify the distinction between lines (theory) and markers (measurements) in the caption.

Point-by-point responses to the reviewers' comments

Reply to the Reviewer #1

Reviewer's Comments: *In the paper "Hybrid topological photonic crystals" by Y. Wang et al. the authors demonstrate that photonic phases with different topologies, namely quantum anomalous Hall and valley Hall phases, can exist in the same photonic crystal. This is so because the quantum anomalous Hall phase requires the breaking of time-reversal (\mathcal{T}) symmetry whereas the valley Hall phase requires the breaking of the parity (\mathcal{P}) symmetry. Therefore, it is no surprising at all that in a system in which both these symmetries are broken both topological phases can be created and as such the main result demonstrated by the authors is expected and to some extent trivial.*

Our Reply: Indeed, in principle, it is possible to have quantum anomalous Hall and valley Hall phases coexist in a single system with both \mathcal{P} and \mathcal{T} symmetries broken. However, in reality, this phenomenon has *never* been realized in experiments, *nor* even has it been proposed in any materials. It is thus very meaningful to demonstrate this in experiments. Furthermore, in the literature considerable efforts have been devoted to exploring photonic systems with multiple topological band gaps owing to their prospective applications. For example, Lan *et al.* demonstrated that nonlinear optical processes, such as second- and third- harmonic generation, can be implemented in a well-designed topological photonic crystal with multiple Chern gaps, which may serve as a new approach to the topology-protected frequency mixing processes in photonics [Phys. Rev. B 101, 155422 (2020)]. Chen *et al.* designed a valley-Hall photonic topological insulator with dual-band valley-Hall topological kink states, which may pave the way for multichannel substate-integrated photonic devices [Adv. Opt. Mater. 5, 201900036 (2019)].

However, the existing works focus mainly on multi-gap topological photonic systems where these topological band gaps are of the same class (e.g., valley Hall band gaps), whereas the intriguing possibility that these topological band gaps can be of distinct classes are not yet explored.

A couple of works have been done on the coexistence of valley and pseudo-spin topological photonic states, which seem to be hybrid topological photonic systems [Phys. Rev. Res. 2, 043148 (2020), Photon. Res. 10, 999 (2022)]. However, these systems are *not* exactly hybrid topological systems because the photonic quantum spin Hall states protected by the concurrent parity (\mathcal{P}) and time-reversal (\mathcal{T}) symmetries is incompatible with the photonic valley Hall states that requires the breaking of \mathcal{P} . The only possible route to hybrid topological photonic systems is to find out a physical setup that hosts concurrently quantum anomalous Hall states and valley Hall states in different band gaps, because these two topological states are compatible with each other in the required symmetries. Hence, our experimental realization of hybrid topological photonic crystals fills this gap and is thus highly valuable in this frontier of topological photonics (as explained more clearly in the response below). The other two reviewers' positive assessments also confirm the importance of our work.

In the revised manuscript, we emphasize accordingly the novelty of our work by highlighting the difference between our work and earlier work in the Introduction part in a more transparent way.

Reviewer's Comments: *Moreover, the authors do not explain why it would be important to have both topological phases in the same photonic crystal. This is an important issue because the photonic system considered by the authors is linear and therefore the two phases with different topologies (topological edge modes) do not couple and therefore they do not interact. As such, the fact that they occur simultaneously in the same photonic system has no practical relevance. It would have been much more interesting to see that the authors consider the case of a nonlinear photonic crystal, so that the optical modes with different topologies can interact.*

Our Reply: We thank the reviewer for this helpful comment that really improves our work. Indeed, in the previous manuscript, we mainly focus on linear properties where the chiral and valley edge modes in different band gaps do not couple with each other. Nevertheless, even for linear properties, from an application perspective, multiplexing edge states with distinct frequencies and topological properties open new possibilities

in the manipulation of photons which may be useful in realizing photonic devices with multifrequency communication channels like frequency division multiplexing that are highly desirable in the integrated photonics.

In addition to the above, **two important results** are added to discussion part of the revised manuscript and the Supplementary Information:

- (i) The intriguing interplay between nonlinear effects and the multi-gap topology is studied where unconventional wave dynamics along the edge boundaries is found.
- (ii) We generalize the multi-gap hybrid topology to Floquet topological insulators by designing a honeycomb Floquet lattice system which exhibits concurrent Chern and valley Hall topology in different quasi-energy gaps (see results in Fig. R3). Since Floquet photonic topological insulators have been realized in various coupled optical waveguide systems at optical frequencies [see, e.g., Nature **496**, 196–200 (2013)], our design opens the possibility of realizing photonic multi-gap hybrid topological phenomena at optical frequencies.

Therefore, in response to the reviewer's comments, we do find interesting nonlinear effects based on the multi-gap hybrid topology which has not seen before. In addition, using Floquet photonic topological insulator design we can generalize our study to optical frequency domain. These findings further promote the importance of our work. We now elaborate on these two results:

First, in the revised manuscript, we added the discussion on the nonlinear effects through which photons in the chiral edge states can be converted to the valley edge states, leading to the interesting interplay between topological phenomena and nonlinear effects. To demonstrate this intriguing effect, we choose a set of parameters to ensure that the frequency of the valley edge states is nearly twice of the frequency of the chiral edge states. As shown in Figs. R1a and R1d, there are two photonic band gaps which satisfy this frequency doubling relation and have distinct topological properties, i.e., quantum anomalous Hall and valley Hall topological band gaps (labeled by the blue and orange regions, respectively). Introducing the nonlinear effect via the second-harmonic generation in the system will enable the coupling between the chiral edge

states in the lower band gap and the valley edge states in the upper band gap. For simplicity, we denote the quantum anomalous Hall gap as gap L and the valley Hall topological gap as gap U.

It can be calculated through Berry curvatures or symmetry indicators that gap L has a Chern number 1 while gap U is a valley Hall topological band gap (see revised Supplementary Note 6). The edge spectra in gaps L and U are presented in Figs. R1b and R1e, respectively, where the electromagnetic wavefunctions are shown in Figs. R1c and R1f, separately, for the edge states with frequency 9.73GHz and 19.46GHz, i.e., these edge states satisfy the frequency doubling relation.

Figure R1 | Hybrid topological photonic crystal with distinct topology in different band gaps. a, d, Bulk bands with a Chern gap and a valley Hall gap, respectively. b, e, The simulated edge dispersions corresponding to the topological band gaps in a and d, respectively. c, f The electric field patterns of the chiral edge states at frequency of 9.73GHz (marked by A in b), and the unbalanced valley edge states at frequency of 19.46GHz (marked by B and C in e).

If the nonlinear effect is turned on, there will be frequency conversion processes via the edge modes as indicated in Fig. R1c and R1f. The full-wave dynamics of the

nonlinearly coupled chiral edge states and valley edge states is calculated numerically using the software COMSOL MULTIPHYSICS (see the revised Supplementary Note 6 for details). The main results are summarized in Fig. R2. In Fig. R2a, we show the excitation of the chiral edge states at the frequency 9.73GHz by a point source via linear properties. In Fig. R2b, we present the excitation of the valley edge states at the frequency 19.46GHz by another point source via linear photonics. In Fig. R2c, we show that when the nonlinear effect is turned on, the valley edge states can be excited by a point source at the frequency 9.73GHz. The nonlinear effect leads to the up conversion from the chiral edge states to the valley edge states (and vice versa) and thus the mixing of the two edge states in different band gaps. Interestingly, without the nonlinear effect, the wave excited at 9.73GHz can only propagate from left to right. By turning on the nonlinear effect, the edge waves can propagate in the opposite direction as well, since the valley edge states are not unidirectional. This feature is entirely different from the results in the literature [Phys. Rev. B 101, 155422 (2020)] where all edge states are propagating in the same direction. The property shown in Fig. R2c can be used to design a porter which can simultaneously convert the frequency of photons and switch its propagation direction in the edge channels.

Figure R2 | Nonlinear frequency conversion between the chiral edge states and valley Hall

edge states. a, Simulated electromagnetic wave pattern excited by a point source with 9.73GHz (labeled by the green star) in the linear photonics domain (i.e., exciting the chiral edge states). **b,** Simulated electromagnetic wave pattern excited by a point source with 19.46GHz (labeled by the blue star) in the linear photonics domain (i.e., exciting the valley edge states). **c,** Simulated electromagnetic wave pattern excited by a point source with 9.73 GHz (labeled by the green star) when the nonlinear effect is turned on.

Reviewer's Comments: *Finally, another serious drawback of the paper is that the physics of the new structure is demonstrated at microwave frequencies, using magnetic materials. Therefore, these results cannot be easily extended to optical frequencies, and as such the results reported here would have very little practical impact.*

Our Reply: In the current manuscript, the hybrid topological system is demonstrated by using gyromagnetic materials, which have weak magneto-optical responsive in the optical frequencies. However, we remark that the results still can be extended to optical frequencies since the design principle of the hybrid topological system merely requires for the breaking of both \mathcal{P} and \mathcal{T} symmetries. In response to the reviewer's comment, we implement the Floquet engineering to realize the hybrid topology in coupled waveguide arrays, which offers a possible scheme to extend to optical frequencies.

The time varying model is shown in Fig. R3a, which contains six steps. The time-dependent Hamiltonian of the system can be described by,

$$H_{\mathbf{k}}(t) = \sum_{m=1,3,5} \theta(t)(e^{ib_m \cdot \mathbf{k}} \sigma^+ + \text{H. c.}) + \Delta(t) \sigma_z, \quad (\text{R1})$$

where $\theta(t)$ is set to be θ at odd steps, and zero at even steps; $\Delta(t)$ equals Δ at even steps, and zero at odd steps; $\sigma^\pm = (\sigma_x \pm i\sigma_y)/2$, where $\sigma_{x,y,z}$ are Pauli matrices; the vectors b_{2n+1} are given by $b_1 = \left(\frac{a}{\sqrt{3}}, 0\right)$, $b_2 = \left(-\frac{a}{2\sqrt{3}}, -\frac{a}{2}\right)$, $b_3 = \left(-\frac{a}{2\sqrt{3}}, \frac{a}{2}\right)$, where a is the lattice constant. The spectrum of the system can be obtained by solving eigen equation of Floquet operator,

$$U_F |\phi\rangle = e^{-i\varepsilon T} |\phi\rangle, \quad (\text{R2})$$

here the Floquet operator is defined as $U_F \equiv \mathcal{T} \exp \left[-i \int_{t_0}^{t_0+T} H(\tau) d\tau \right]$, where \mathcal{T} is

the time-ordering operator. Different from the static system, here we study the quasi-energy ε , which has a periodicity of $2\pi/T$.

From the model, we notice the time reversal symmetry of the system is broken due to the periodically driving and the parity symmetry is also broken when Δ is nonzero. Besides, the lattice sites are arranged in a honeycomb lattice so that the model is suitable for studying valley topology and quantum anomalous Hall physics, and may support hybrid topological states. The phase diagram of the system is shown in Fig. R4d, which contains topological trivial state with vanishing Chern number, Chern insulator with Chern number ± 1 and anomalous Floquet topological insulator also with vanishing Chern number but supports chiral edge state. We study the strip structure shown in Fig. R3b which contains the interface between topological trivial state (upper) and hybrid topological state (lower) with different valley topology. The Chern number is -1 for the first bulk band of HTPC which can be obtained from Berry phase calculation in Fig. R4b, we can see the winding of Berry phase as function of k_1 is -1 . The Chern number is 0 for topological trivial state whose Berry phase winding as function of k_1 is 0 as shown in Fig. R4c. The quasi-energy spectrum of the strip structure is shown in Fig. R3c. There are two bulk bands due to the bipartite nature of the system and two band gaps, one at quasi energy 0 and the other at quasi energy π/T . The π/T gap belongs to quantum anomalous Hall phases and supports chiral edge state and chiral interface states colored by red in Fig. R3c. The 0 gap of topological trivial state and HTPC state belongs to different quantum valley Hall phases (the sign of Δ is opposite) so that there are valley interface states colored by blue. The field distribution of chiral edge states, chiral interface state and valley interface states are shown in Fig. R3d. We notice the field of the chiral edge state is localized at the lower edge which can be treated as the interface between HTPC and vacuum, the fields of the chiral interface state and valley interface state are localized at interface. It is worthy to mention that, although here we propose a time varying system, in the real optical experiment we can use the coupled helical waveguides arrays which treat one spatial direction as time. Such system has been used to study photonic Floquet topological insulator [Nature 496, 196 (2013); Nat. Mater. 19, 855 (2020)], photonic valley topological insulator [Phys. Rev. Lett. 120,

063902 (2018)], photonic anomalous Floquet topological insulator [Phys. Rev. Lett. 117, 013902 (2016); Nat. Commun. 8, 13756 (2017); Nat. Commun. 8, 13918 (2017); Nat. Mater. 21, 634 (2022)] and photonic anomalous Floquet higher-order topological insulator [Phys. Rev. B 103, L041402 (2021)] in optical region. Our proposal here provides a new member to the family of topological insulator in optical region.

Overall, we believe that with the above response and revisions, our manuscript will be found suitable for publication in Nature Communications.

Figure R3 | Hybrid topology in coupled waveguide arrays system via Floquet engineering. a,

the time varying model which is composed of six steps. In odd steps the couplings strength is θ and in even steps the on-site potential difference is Δ . **b**, The strip structure with an interface between topological trivial state and HTPC state with different valley topology. The topological trivial state with $\theta = 0.16\pi$ and $\Delta = 0.16\pi$ has vanishing Chern number for both bands and the Chern number of HTPC state with $\theta = 0.25\pi$ and $\Delta = 0.33\pi$ is ± 1 for two bands. **c**, The quasi energy spectrum of the strip structure. The chiral edge states are colored by blue and the valley edge states are colored by red. **d**, The field distribution of chiral edge states and valley edge states marked in **(c)**.

Figure R4 | Numerical calculation of Chern numbers. **a**, Brillouin zone of the honeycomb lattice. **b**, **c**, The Berry phase as a function of k_1 for different bands. The parameters are $\theta = 0.25\pi, \Delta = 0.33\pi$ for **b** and $\theta = 0.16\pi, \Delta = -0.16\pi$ for **c**. The Chern numbers are -1 and 0 for **b** and **c** respectively. **d**, Phase diagram. Chern numbers are marked. AFI is the anomalous Floquet topological insulator.

Reply to the Reviewer #2

Reviewer's Comments: *In the manuscript by Yanan Wang, et al., the authors propose and experimentally investigate a hybrid topological photonic system which simultaneously exhibits anomalous Hall effect (AHE) type of topology with broken time-reversal symmetry and valley-Hall (VH) type of topology with broken inversion symmetry. As such, the system is characterized by two topological invariants, and allows multiplexing via two kinds of topological boundary states. To the best of my knowledge, this is first experimental study which would show such multispectral robust topological transport via different topological states in different band gaps. The paper is also very well-written, experiments are fascinating, and I only suggest to slightly tune down hype in the abstract (e.g. "rich photonic edge transport measurements", which sounds really too much since measurements cannot be rich).*

Our Reply: We thank the reviewer for his/her appreciations of our work. According to the reviewer's comment, we change the "rich photonic edge transport measurement" to "photonic edge transport measurement".

Reviewer's Comments: *Another point of criticism I would like to raise is that the authors do not cite earlier works which too studies hybrid topological systems, including combining AHE and VH effects (theoretical work by Ni et al., Science Advances, 7, 2, (2021)), as well as another experimental work combining spin-Hall (SH) and VH (Kang et al., Nature Commun. 9, 3029 (2018)) based on earlier SHE experimental work (Cheng et al., Nature Materials 15, 542 (2016)). Both works study the effect of two mass terms, topological transitions between different topological phases, and evaluate proper hybrid topological invariants. I note, however, that these earlier works, while very relevant, do not undermine novelty of the manuscript under review, because here heterogeneous modes in topologically distinct and spectrally separated bandgaps are studies for the first time. Nonetheless, earlier works must be given proper credit in the manuscript with appropriate mentions and citations.*

Our Reply: We thank the reviewer for bringing these references to us. In the revised

manuscript, we cited these papers in the introduction and in places wherever they are relevant.

Reviewer's Comments: *Provided these minor suggestions are implemented, I strongly recommend this work for publication in Nature Communications. I think this is an important result which shows many novel aspects which can be achieved in hybrid topological photonic systems, including higher values of hybrid topological invariants, heterogeneous multispectral topological transport, etc. I am confident that this paper will be of significant interest to the broad research community and Nature Communications readership.*

Our Reply: We thank the reviewer for his/her recommendation of our work for publication in Nature Communications.

Reply to the Reviewer #3

Reviewer's Comments: *The manuscript 'Hybrid topological photonic crystals' by Y. Wang, H.-X. Wang, et al. reports theoretical designs and experimental demonstrations of a 2D T- and P-broken photonic crystal featuring gaps of distinct nontrivial topology, namely Chern and valley Hall insulating gaps. The design concept and approach is interesting and clearly displays the intended phenomenon. Similarly, the experimental demonstrations in the microwave-regime back up the theoretical predictions very convincingly. Even separately for Chern and valley Hall edge states, the demonstrations are among the clearest that I've seen. Finally, the authors suggest a possible routing application for their design. The work and methodology is sound and explained clearly.*

Our Reply: We thank that the reviewer for his/her careful reading and appreciations of our work.

Reviewer's Comments: *I am unsure how impactful the idea of a dual-gap-topology ultimately could be in practice, especially of valley Hall type topology. Nevertheless, the demonstration is beautiful and amounts to a particularly clear illustration of the potential of "abundant" co-existing topological phenomena in photonic crystals. The design idea is also neat— particularly, in using a structural rotation angle to realize the desired dual-gap topology. Overall, despite my doubts about the impact and practical utility of dual-gap-topology, I believe the clarity of the experimental results and presentational quality is sufficiently high that the paper will have clear impact in the community of topological photonics. Provided the authors can fix or reply to the few points I raise below, I would recommend publication in Nature Communications.*

Our Reply: We thank the reviewer for his/her appreciations and recommendation of our work. Regarding the innovation and value of this work, we emphasize the following: First, realizing multi-gap topology of distinct class is an important progress in future topological photonics in the nonlinear regime. As shown in Fig. R2, the nonlinear coupling between the chiral edge states in the lower band gap and the valley edge states

in the higher band gap enables simultaneous tuning of the frequency and guiding of the photon propagation direction at the edge boundary (in a sense, it serves as a nonlinear switch of photonic flows in the edge channels). This can be used to design novel functions for topological integrated photonic circuits. Second, even without nonlinear couplings, the discovery here enriches multiplexing of topological wave guiding of photons in multi-frequency-bands with distinct properties which may provide more flexibility in manipulating electromagnetic waves with promising potentials for multiband and multifunctional applications. Third, with the generalization of the hybrid topology to Floquet photonic topological insulators, our work can inspire future exploration of various kinds of Floquet photonic topological insulators. With these impacts, we believe that the studies here can be very useful and impactful for future research in photonics.

Reviewer's Comments: *Main points:- The authors claim in the introduction (and, to a degree, in the abstract) that 'It is unclear whether photonic band gaps in a single system can have distinct topologies'. Technically, this is true in that no one has explicitly shown this - but conceptually, it is obviously possible. Of course, the existence of practical designs is not a given. Nevertheless, I suggest the authors temper this statement to reflect that the real uncertainty here is not whether it is possible to have distinct band topologies in distinct gaps, but just whether practical designs of such dual-gap topology can exist and how we should think about engineering them.*

Our Reply: We thank *the* reviewer for the helpful comments and suggestions. We revised the introduction accordingly. Explicitly, the introduction is revised as follows:

“However, most studies focus on multi-gap topological photonic systems with the same topological class^{38,47,49,50}, while that with distinct topological classes has not yet been realized. It was reported that the topological valley and pseudo-spin edge states can coexist in a Kekulé photonic system⁴⁸ and a composite photonic crystal⁵¹, which seem to be multi-gap topological photonic systems with different topological classes. However, from the symmetry consideration of photonic topological phases in two dimensions, the photonic quantum spin Hall insulator

phase is protected by the concurrent parity (\mathcal{P}) and time-reversal (\mathcal{T}) symmetries. In comparison, the photonic VH insulator phase requires the breaking of \mathcal{P} , while the photonic QAH insulator phase requires the breaking of \mathcal{T} . Therefore, the photonic quantum spin Hall insulator phase is incompatible with the latter two. In contrast, the photonic VH insulator and QAH insulator phases are compatible with each other. Therefore, it is possible in principle to have multi-band topology of the QAH and VH types in a single photonic system if both \mathcal{P} and \mathcal{T} are broken which, however, has not yet been realized.”

Reviewer’s Comments: - *Can the authors comment on the near-constant frequency shift between theory and experiments, especially pronounced for the valley edge state (e.g., Fig. 3b and Supplementary Fig. 6)?*

Our Reply: In the experimental measurements, it is unavoidable to lift the upper cladding layer and introduce an air gap between the photonic crystal and the upper cladding layer to enable near-field scanning of the electromagnetic waves. This air gap has non-negligible effect on the photonic bands. Depending on the frequency of photons, this effect can be small (for low frequency photons which have wavelength much larger than the air gap) or considerable (for high frequency photons). In a small frequency range, the air gap causes constant frequency shift in the photonic band structure which is the origin of the near-constant frequency shift between theory and experiments. This effect, however, is very difficult to be reproduced in simulation (because essentially the simulation will be a 3D problem). Nevertheless, this near-constant frequency shift does not change the basic features that we are interested, i.e., the main features of the band structure, the band topology, and the dispersion of the edge states. These main features are all reproduced in experiments in our study. In response to this comment, we added the above explanations to the Supplementary Note 5.

Reviewer’s Comments: - *The authors note a selection of earlier work [43-48] on dual band topology but do not give a clear exposition of how they differ from the present work. Can the authors expand on these differences, e.g. in the Introduction? For*

instance, [44-47] are suggested to have identical topology in distinct bands - in what sense are they then dual band topology? Perhaps especially relevant is a clarification relative to Ref. 48.

Our Reply: In response to this comment, we point out the difference between the earlier work and our work.

In the revised manuscript, we highlight that “most studies focus on multi-gap topological photonic systems with the same topological class^{38,47,49,50}, while that with distinct topological classes has not yet been realized. It was reported that the topological valley and pseudo-spin edge states can coexist in a Kekulé photonic system⁴⁸ and a composite photonic crystal⁵¹, which seem to be multi-gap topological photonic systems with different topological classes.”

Reviewer’s Comments: *Minor points/suggestions: * Figure 1b,c,e: to facilitate the discussion of how the Chern number of band 4 changes under variations of θ , the authors should label what the Chern number is of band 5. Since the change of Chern number of band 4 is achieved by an inversion with band 5, the sum of Chern numbers of the two should be invariant wrt. θ before after 9.5° . On this point, the Chern number of band 4 changes by 2 (from 1 to -1) under the band inversion: nominally, this is surprising, since a single band inversion usually only contributes a change of 1. The authors should explain why this is possible (is it e.g., related to the unpaired nature of the quadratic degeneracy at 9.5° ?).*

Our Reply: Since there exists a band crossing between band 5 and 6 at $\theta = 0^\circ$, therefore, it is impossible to calculate the Chern number of band 5 separately. Instead, we calculated the Chern number of band 5 with $\theta = 30^\circ$ and $\theta = 8^\circ$, respectively, which represent to the two cases of before and after 9.5° . The results show that the Chern number of band 5 at $\theta = 30^\circ$ is 0, while that at $\theta = 8^\circ$ is -2.

Indeed, the Chern number of band 4 changed by 2 under the band inversion. Such a band inversion can be understood via the $\vec{k} \cdot \vec{p}$ Hamiltonian of the unpaired quadratic Dirac point, which gives

$$H(\mathbf{k}) = \mathbf{h}(\mathbf{k}) \cdot \boldsymbol{\sigma} = v_D [(k_x^2 - k_y^2)\sigma_x - 2k_x k_y \sigma_y] + M\sigma_z, \quad (\text{R3})$$

where v_D is the Dirac velocity, $\mathbf{k} = (k_x, k_y)$ measures the momentum deviation from the K valley, M refers to the mass term induced by the C_{3v} and/or T symmetries, and $\sigma_i (i = x, y, z)$ are Pauli matrices operating on the orbital degree of freedom. The Berry curvature is then given by

$$\Omega(\mathbf{k}) = \frac{\mathbf{h}}{2\hbar^3} \cdot (\partial_{k_x} \mathbf{h} \times \partial_{k_y} \mathbf{h}) = \frac{-2Mv_D^2 \mathbf{k}^2}{[v_D^2 \mathbf{k}^4 + M^2]^{\frac{3}{2}}}. \quad (\text{R4})$$

The valley Chern number can be attained by integrating the Berry curvature over the whole 2D space, which finally gives $C_K = -\text{sgn}(M)$. Hence, accompanying the closing and reopening of the unpaired quadratic Dirac point, the mass term M changes its sign and contributes to a change of 2 during the band inversion process.

Reviewer's Comments: * *Figure 1(a-b): Although the figures are conceptual and hence do not require explicit frequency units or values, it would be helpful if the authors could include relevant labels on the momentum axis (since the valley Hall effect involves a specific choice of K or K' valley); even just an indication of the zero-momentum would be helpful.*

Our Reply: In response to this comment, we add Γ , K and K' valleys into the momentum axis. The replaced figures are listed as follows.

Reviewer's Comments: * 'Unbalanced valley edge states' are mentioned a few times without immediate explanation. I suggest the authors either give a brief description of what is meant by 'unbalanced' at the first mention, or hold off on introducing the modifier until it is explained. (As I understand it from the text, different valleys get different absolute group velocities)

Our Reply: The reviewer has a correct understanding of the “unbalanced valley edge states”. In response to this comment, we add a brief description of the unbalanced valley states in the caption of Fig.1 as follows:

“The valley Hall gap hosts unbalanced valley edge states where the absolute group velocity of the valley edge state around the K valley is different from that around the K' valley.”

Reviewer's Comments: * p. 4, bottom: The authors mention 'paired Dirac points': without reference to the SI, I could not guess that this meant a 'doublet of Dirac points', one at K and K' (as required by time-reversal symmetry). I suggest the authors be slightly more elaborate here.

Our Reply: In response to this comment, we revised the “paired Dirac points” as follow:

“...the band structure has some paired Dirac points at $K(K')$ and a quadratic point at Γ that are protected by both C_{3v} and \mathcal{T} symmetries (see Supplementary

Fig. 1)”.

Reviewer’s Comments: * Eq. (1): The first two terms ostensibly feature an implicit Kronecker product with τ_0 : this should be made explicit for clarity's sake.

Our Reply: In response to this comment, we revise Eq. (1) as follow:

$$H(\vec{k}) = A_Q \hat{\tau}_0 [(k_x^2 - k_y^2) \hat{\sigma}_x - 2k_x k_y \hat{\sigma}_z] + B_Q k^2 + (m_T \hat{\tau}_0 - m_V \hat{\tau}_z) \hat{\sigma}_z$$

Reviewer’s Comments: * The authors refer to 'boundary-configured valley edge states' and reference [57,58]. I don't believe this is widely known terminology: a brief explanation of the term's meaning is appropriate here.

Our Reply: In response to this comment, we add a brief explanation in the context as following:

“Furthermore, there is no edge state in the lower edge boundary in gap III since the HTPC with rigid boundary does not strictly contribute a bulk-edge correspondence and the edge dispersions also depend on the boundary configuration, making it boundary-configurable valley edge states^{60,61} (see Supplementary Note 3 for more details)”.

Reviewer’s Comments: * Fig. 3b: it would be nice to clarify the distinction between lines (theory) and markers (measurements) in the caption.

Our Reply: In response to this comment, we revise the caption of Fig. 3b as following:

“The simulated (solid lines) and experimental (empty circles) edge dispersions.”

REVIEWER COMMENTS

Reviewer #1 (Remarks to the Author):

I have carefully read the Authors' response to my comment and my opinion about the paper is unchanged - it should not be published in Nature Communications. The authors simply combine two well understood phenomena and demonstrate them in the same device. Therefore, there is no original idea in this paper and, in fact, should not have been sent out for review. The fact that the experiments are properly done is not impressive at all - the scope of a journal like Nature Communications should not be to report well-done experiments.

The authors considered my suggestion to study the nonlinear interaction between modes of different nature, but their approach is incorrect. It should be clear that if the material is quadratically nonlinear then some field will be generated at the SH, but this is irrelevant. The key concept here is phase-matching. Without phase matching the nonlinear interaction is inefficient and useless in practice. The authors do not even touch this point. In conclusion, I recommend this paper to be rejected by Nature Communications.

Reviewer #3 (Remarks to the Author):

The authors have made commendable efforts to improve their manuscripts and to reply to the concerns raised in the first review round. The manuscript now makes a more convincing case for the relevance of multiband topology, especially through the incorporation of an interesting nonlinear example. Several other improvements to the manuscript are also evident, most notably in more precise and clearer phrasing and by multiple, significant additions to the Supplementary Information.

Considering the above revisions and improvements, I am happy to recommend publication in Nature Communications.

I made note of a few minor issues that either appeared during revisions or existed before: see below. The authors may consider them at their convenience; I require no replies.

- Eq. (1): I am happy to see a revision of the first term in the equation that makes the τ_0 term explicit. The term $B_Q k^2$ term, however, still has its orbital degrees implicit: I suggest the authors add the term $\tau_0 \sigma_0$ in the interest of clarity and consistency.

- Fig. 1b: Since this is a projected band diagram, unlike Fig. 1a, the k-labels cannot be Γ , K, and K' since they refer to points in the 2D Brillouin zone. If the authors intended to indicate the projections of these points to the 1D Brillouin zone, they could e.g. follow the convention of add a line above each label.

- The authors note in their reply that it is not possible to calculate the Chern number of band 5 at $\theta=0^\circ$ since bands 5 and 6 touch there. I suggest the authors then simply compute the composite Chern number of the multiplet of bands 5 and 6, which remains well-defined. For computation, either the non-Abelian Wilson loop formulation [see e.g., Z2Pack, <https://arxiv.org/abs/1610.08983>] or the multiband Chern formulation (see e.g., <http://dx.doi.org/10.1143/JPSJ.74.1674>) can be used.

- The code availability section contains the sentence "We use the commercial software COMSOL MULTIPHYSICS to perform the `_acoustic_` wave simulations [...]": This seems to be a copy-paste typo: did the authors mean "electromagnetic wave simulations" rather than "acoustic"?

Point-by-point responses to the reviewers' comments

Reply to the Reviewer #1

Reviewer's Comments: *I have carefully read the Authors' response to my comment and my opinion about the paper is unchanged - it should not be published in Nature Communications. The authors simply combine two well understood phenomena and demonstrate them in the same device. Therefore, there is no original idea in this paper and, in fact, should not have been sent out for review. The fact that the experiments are properly done is not impressive at all - the scope of a journal like Nature Communications should not be to report well-done experiments.*

Our Reply: We disagree with the reviewer. Our innovation is sitting at the firm ground of proposing and observing a hybrid topological photonic system that has *never* been studied before. In particular, we achieved for the first time the experimental access to two distinct types of topological edge phenomena in a single 2D photonic system. We further discuss the possible physical consequences when the distinct topological edge states are coupled together through nonlinear optical effects. In this way, we find novel functions that can enable the switch of the photon energy flow in the topological edge channels via nonlinear effects which has *never* been thought to be possible. We went on to show that hybrid topological phenomena can also appear in optical frequencies in Floquet photonic topological insulators. For such a purpose, we even propose a concrete design of a novel Floquet photonic topological insulator with multi-gap hybrid topology. The underlying physics of the rich hybrid topological phenomena is also discussed in depth. It is worth mentioning that both the photonic quantum anomalous Hall insulator gap and the photonic valley Hall gap studied here are distinct from the ones known in the literature and we have created new theories to describe them. We strongly believe that these innovations contribute significantly to the study of topological phenomena and the field of topological photonics and will thus be impactful and inspiring in these research frontiers.

Reviewer's Comments: *The authors considered my suggestion to study the nonlinear interaction between modes of different nature, but their approach is incorrect. It should be clear that if the material is quadratically nonlinear then some field will be generated at the SH, but this is irrelevant. The key concept here is phase-matching. Without phase matching the nonlinear interaction is inefficient and useless in practice. The authors do not even touch this point.*

Our Reply: In the previous report of the reviewer, the phase matching was not mentioned. The major issue there is whether the nonlinear optical effects can induce any interesting phenomenon on the edges, we thus did not go into the details on the phase-matching. Here, we can address this issue with a positive response, i.e., the phase-matching is indeed important and by tailoring the dispersion of the edge states we can achieve the quasi-phase-matching condition to enhance the nonlinear optical effect on the manipulation of the edge propagation of photons. For this purpose, we tailor the edge dispersions by cutting part of the unit-cells at the edge boundary. This is illustrated in Fig. R1b with a geometry parameter $p = 0.4a$ where $a = 21\text{mm}$ is the lattice constant. The resultant dispersion of the topological edge states in the quantum anomalous Hall photonic band gap around 10GHz is shown in Fig. R1a, while the electric field profile of a specific edge state (labeled the red-letter A) is shown in Fig. R1c. The dispersion of the edge states in the valley Hall photonic band gap is shown in Fig. R1d.

Figure R1 | Hybrid topological photonic crystal with edge states satisfying frequency and phase matching for second-harmonic generation. **a** and **d**, The projected band structures at different frequency ranges. The red and blue curves correspond to the edge states for the upper and lower edge boundary of the supercell model in **b**, respectively. Here, **A** denotes an edge state in the lower band gap with a frequency $f_0 = 10.07\text{GHz}$, while **B** and **C** denote two edge states in the higher band gap with a frequency $f_2 = 2f_0 = 20.14\text{GHz}$. **b**, the schematic of the model with the upper edge termination is truncated at $p = 0.4a$. **c, e, f**, are the electric field distributions of the edge states **A**, **B**, and **C**, separately. This figure is included in the supplementary information (SI) as new supplementary figure 10.

We consider here the second-harmonic generation between the edge state **A** in Figs. R1a and R1c with the fundamental frequency $f_0 = 10.07\text{GHz}$ and the wavevector $k_0 = 0.495 \frac{\pi}{a}$ and the resonant second harmonic edge states **B** and **C** in Figs. R1d, R1e, and R1f with a frequency $f_2 = 20.14\text{GHz}$. The edge states **B** and **C** have the wavevectors $k_{21} = 0.935 \frac{\pi}{a}$ and $k_{22} = \frac{\pi}{a}$, respectively. Interestingly, the edge states **A** and **B** have positive group velocity, while the edge state **C** has negative group velocity. The wavevector mismatch for the edge states **B** and **C** are small, $\Delta k_1 = k_{21} - 2k_0 = -0.055 \frac{\pi}{a}$ and $\Delta k_2 = k_{22} - 2k_0 = 0.01 \frac{\pi}{a}$, both showing quasi-phase-matching.

The edge state A can be visualized by a point source excitation at the edge boundary with the frequency f_0 (see Fig. R2a), showing elegantly the unidirectional photonic edge propagation (right-going). On the other hand, the edge states B and C can both be excited by a point source at the same position with frequency f_2 (Fig. R2b), showing bidirectional photonic edge propagation with asymmetric features (i.e., the right-going signal decays faster than the left-going signal). With second-harmonic nonlinear effects and exciting at the frequency f_0 at the same position, the photonic energy flow along the edge channel now switches from unidirectional to bidirectional (Fig. R2c). The simulation results in Fig. R2c show comparable left-going and right-going photonic energy flow which is an evidence that the second-harmonic frequency conversion is efficient. We remark that the above phenomenon has not yet been found before, since in the previous studies, the topological edge states in different band gaps propagate in the same direction [Phys. Rev. B 101, 155422 (2020)].

With the above calculations and discussions, which have been added to supplementary Note 6 in the revised supplementary information, we demonstrate through a concrete example that the phase-matching condition can be satisfied to achieve efficient second-harmonic frequency conversion among the edge states in distinct topological band gaps. Such frequency conversion can be used to manipulate photonic energy flows in the topological edge channel which is a new degree of freedom that can be promising for future topological photonics. We believe that these results are significant enough to attract readers attention. We hope the above discussions and revisions are satisfactory to the reviewer.

Figure R2 | Efficient second harmonic frequency conversion in the edge channel of the hybrid topological photonic crystal. a, b, (Without nonlinear optical effects) Simulated electrical field distributions for the cases with a point source excitation at the frequency $f_0 = 10.07\text{GHz}$ (source labeled by the green star) and $f_2 = 20.14\text{GHz}$ (source labeled by the blue star), respectively. **c,** (With second-harmonic nonlinear optical effects) Simulated electrical field distribution for the case with a point source excitation at the frequency $f_0 = 10.07\text{GHz}$. This figure is included in the revised supplementary information as new supplementary figure 11

Reply to the Reviewer #3

Reviewer's Comments: *The authors have made commendable efforts to improve their manuscripts and to reply to the concerns raised in the first review round. The manuscript now makes a more convincing case for the relevance of multiband topology, especially through the incorporation of an interesting nonlinear example. Several other improvements to the manuscript are also evident, most notably in more precise and clearer phrasing and by multiple, significant additions to the Supplementary Information. Considering the above revisions and improvements, I am happy to recommend publication in Nature Communications. I made note of a few minor issues that either appeared during revisions or existed before: see below. The authors may consider them at their convenience; I require no replies.*

Our Reply: We thank the reviewer for his/her appreciation and recommendation of our work.

Reviewer's Comments: - *Eq. (1): I am happy to see a revision of the first term in the equation that makes the τ_0 term explicit. The term $B_Q k^2$ term, however, still has its orbital degrees implicit: I suggest the authors add the term $\tau_0 \sigma_0$ in the interest of clarity and consistency.*

Our Reply: In response to this comment, we revise Eq. (1) as follow:

$$H(\vec{k}) = A_Q \hat{\tau}_0 [(k_x^2 - k_y^2) \hat{\sigma}_x - 2k_x k_y \hat{\sigma}_z] + B_Q k^2 \hat{\tau}_0 \hat{\sigma}_0 + (m_T \hat{\tau}_0 - m_V \hat{\tau}_z) \hat{\sigma}_z$$

Reviewer's Comments: - *Fig. 1b: Since this is a projected band diagram, unlike Fig. 1a, the k -labels cannot be Γ , K , and K' since they refer to points in the 2D Brillouin zone. If the authors intended to indicate the projections of these points to the 1D Brillouin zone, they could e.g. follow the convention of add a line above each label.*

Our Reply: In response to this comment, we add Γ , K and K' valleys into the momentum axis. The replaced figures are listed as follows.

Reviewer's Comments: - *The authors note in their reply that it is not possible to calculate the Chern number of band 5 at $\theta=0^\circ$ since bands 5 and 6 touch there. I suggest the authors then simply compute the composite Chern number of the multiplet of bands 5 and 6, which remains well-defined. For computation, either the non-Abelian Wilson loop formulation [see e.g., Z2Pack, <https://arxiv.org/abs/1610.08983>] or the multiband Chern formulation (see e.g., <http://dx.doi.org/10.1143/JPSJ.74.1674>) can be used.*

Our Reply: We thank the reviewer for bringing these references to us. We recalculate the Chern number of the multiplet of bands 5 and 6 via the non-Abelian Wilson loop approach. Since the integral of the Berry connect gives the Berry phase, we can divide the Brillouin zone into many small segments (e.g., N_k) and approximating the integral as the summation of the contributions from each small segment. For the situation of band set (e.g., containing N bands), the Berry phase for the segment is a matrix $\hat{M}^{k_i, k_{i+1}}$, of which the matrix elements are given by

$$M_{nn'}^{k_i, k_{i+1}} = \langle u_{n, k_i} | u_{n', k_{i+1}} \rangle, \quad n, n' \in 1, 2, \dots, N, \quad (R1)$$

where $u_{n, k}$ refers to the periodic part of Bloch wavefunction of n th band at k wavevector. The Berry phase for a loop in Brillouin zone can be obtained by the matrix

product of $\widehat{M}^{k_i, k_{i+1}}$ in the loop through the

$$\widehat{W} = \prod_{i=1}^{N_k} \widehat{M}^{k_i, k_{i+1}}, \quad (R2)$$

The eigenvalues of the above Berry phase matrix contain the information on the Berry of multiplet bands, which is given by

$$\theta = i \log(w_n), \quad n = 1, \dots, N. \quad (R3)$$

In our case, the Bloch wavefunctions are replaced by the electric fields, and the band 5 and 6 are regarded as a band set. Following the above procedures, we calculate the Chern number of the multiplet of bands 5 and 6. As shown in Fig. R3a, we deform the first Brillouin zone from a hexagon to a rhomb. In this way, a closed loop is formed by running from $-\pi$ to π for each given k_1 . The Berry phase $\theta(k_1)$ of the multiplet band of 5 and 6 is given in Fig. R3b, of which winding number gives the Chern number. From Fig. R3b, the Chern number of the multiplet band of 5 and 6 together is -3.

Figure R3 | Hybrid topological photonic crystal with distinct topology in different band gaps.

a, The first Brillouin zone of triangular lattice. **b**, The Berry phase $\theta(k_1)$, which gives the Chern number of multiplet band of 5 and 6.

Reviewer's Comments: - *The code availability section contains the sentence "We use the commercial software COMSOL MULTIPHYSICS to perform the `_acoustic_` wave simulations [...]": This seems to be a copy-paste typo: did the authors mean "electromagnetic wave simulations" rather than "acoustic"?*

Our Reply: We thank the reviewer for his/her careful reading. In response to this comment, we change the “acoustic wave simulations” with the “electromagnetic wave simulations”.

REVIEWERS' COMMENTS

Reviewer #3 (Remarks to the Author):

The authors have satisfactorily resolved my previous comments and I remain supportive of publication in Nature Communications.

(As a minor note, despite the authors calculating the Chern numbers of bands 4, 5, and 6 now in their replies, they still do not include this information in Fig. 2b,c,e; if this is by accident, I encourage them to correct it)than "acoustic"?

Point-by-point responses to the reviewers' comments

Reply to the Reviewer #3

Reviewer's Comments: *The authors have satisfactorily resolved my previous comments and I remain supportive of publication in Nature Communications.*

Our Reply: We thank the reviewer for his/her appreciation and recommendation of our work.

Reviewer's Comments: *(As a minor note, despite the authors calculating the Chern numbers of bands 4, 5, and 6 now in their replies, they still do not include this information in Fig. 2b,c,e; if this is by accident, I encourage them to correct it)*

Our Reply: In response to this comment, we add the Chern number information into the Figure 2b, c, e. The replaced figures are listed as follows.